# Targeting Skin Neoplasms: A Review of Berberine’s Anticancer Properties

**DOI:** 10.3390/cells14141041

**Published:** 2025-07-08

**Authors:** Anna Duda-Madej, Patrycja Lipska, Szymon Viscardi, Hanna Bazan, Jakub Sobieraj

**Affiliations:** 1Department of Microbiology, Faculty of Medicine, Wroclaw Medical University, Chałubińskiego 4, 50-368 Wrocław, Poland; 2Faculty of Medicine, Wroclaw Medical University, Ludwika Pasteura 1, 50-367 Wrocław, Poland; patrycja.lipska@student.umw.edu.pl (P.L.); szymon.viscardi@student.umw.edu.pl (S.V.); hanna.bazan@student.umw.edu.pl (H.B.); jakub.sobieraj@student.umw.edu.pl (J.S.)

**Keywords:** anticancer activity, berberine, melanoma, skin neoplasms, squamous cell carcinoma, natural compounds, adjuvant therapy

## Abstract

Skin cancers are associated with a significant psychological burden across all age groups, particularly as their global incidence continues to rise. Ultraviolet (UV) radiation—primarily UVA and UVB—remains the leading etiological factor, inducing DNA mutations in key genes such as TP53 and BRAF. Among skin cancers, basal cell carcinoma (BCC) is the most prevalent and typically indolent. In contrast, squamous cell carcinoma (SCC) tends to be more invasive, while melanoma is the most aggressive and prone to metastasis. Melanoma is especially concerning due to its rapid dissemination and its occurrence not only on the skin but also in ocular, mucosal, and nail tissues. These challenges, along with rising treatment resistance and mortality, underscore the urgent need for novel anticancer agents. Berberine—a plant-derived isoquinoline alkaloid—has attracted increasing attention for its broad-spectrum anticancer potential, including against skin cancers. In this review, we summarize current evidence regarding berberine’s mechanisms of action in melanoma and SCC, emphasizing both its preventive and therapeutic effects. We further explore its potential as an adjuvant agent in combination with conventional treatments, offering a promising avenue for enhancing the clinical outcomes of skin cancer therapy.

## 1. Introduction

### 1.1. Epidemiology and Characteristics of Skin Cancers

Skin cancer is one of the most common malignancies and a serious threat to global public health. According to WHO data, skin cancer was confirmed in over 1.5 million new cases in 2020, with as many as 325,000 cases being related to melanoma [1]. As reported by the IARC (International Agency for Research on Cancer) report published in 2022, out of all new cases of malignant tumors recorded worldwide (about 20 million), melanoma ranked 17thamong all cancers, with nearly 60,000 patients dying from the disease [2]. It is particularly worrying that the incidence of melanoma is predicted to increase by more than 50% between 2020 and 2040, with the number of deaths related to it reaching approximately 96,000 [1]. In 2020, more than 890,000 new cases of head and neck cancer (squamous cell carcinoma) were diagnosed, with a projected one million cases per year by 2030 [3]. Figure 1 presents the general division of malignant tumors of a skin origin.

Malignant skin tumors can be divided into two groups, melanoma and non-melanoma skin cancers (NMSC), including basal cell carcinoma (BCC) and squamous cell carcinoma (SCC) [4]. Non-melanoma skin cancers, in particular BCC and SCC, account for more than 95% of all skin cancers (70% and 25% of NMSC, respectively) [5]. Histologically, BCC skin cancer, which originates from the basal layer of epithelium, is the most commonly diagnosed skin cancer and has a lower malignancy rate than SCC (75% of NMSC-related deaths) originating from the stratum spinosum [6]. Although melanoma, a malignant tumor of melanocytic origin, is less prevalent, it is still the cause of most skin cancer-related deaths. Figure 2 presents the histological structure of the skin with a mapping of the cancer process in the course of melanoma and NMSC. The main risk factors and characteristics of mentioned neoplasms are presented in Table 1.

### 1.2. Molecular Pathophysiology of Skin Cancers

At the molecular level, the pathogenesis of BCC is (in a majority of cases, 70%) a result of mutation in gene *PTCH1* (patched homolog 1 protein) and *SMO* (gene encoding the Smoothened protein), both of which are involved in Hedgehog signaling pathway [9]. The Hedgehog signaling pathway is not inhibited due to mutations in the PTCH1 gene, which encodes the membrane receptor that, under normal conditions, inhibits the SMO receptor [19,20]. As a result, the constantly active SMO receptor activates GLI family transcription factors (glioma-associated oncogene homolog), which induce the overexpression of effector genes that are key for angiogenesis, proliferation, and the avoidance of apoptosis [21,22]. The molecular basis of the most common mutations in BCC development is summarized in Figure 3.

Mutations in the tp53 gene are the main mechanism of SCC development [23]. The mutations in the *tp53* gene are primarily due to the action of UV radiation, which results in the formation of the so-called pyrimidine dimers and the loss of suppressor gene function [24]. Mutations in the *tp53* gene have numerous implications in the cellular signaling pathways associated in carcinogenesis. The profile of *tp53*-related anti-neoplastic activities is summarized in Figure 4.

Impaired differentiation of keratinocytes occurs due to the loss of suppressor function, similar to tp53, when the *NOTCH* 1-3 genes are inactivated [25]. Mutations in NER (nucleotide excision repair) genes, which are responsible for removing DNA damage caused by factors such as UV radiation, are critical contributors to SCC development [26]. NER dysfunction results in premalignant states such as xeroderma pigmentosum [27]. Additionally, the dysregulation of EGFR-dependent signaling pathways promotes the anti-apoptotic hyperactivation of the PI3K/AKT, JAK/STAT, and ERK/MAPK loops [28]. The RAS (rat sarcoma virus protein) pathway is affected in tumorigenesis, is connected with *tp53* malfunction, and is believed to play a role in SCC development [29]. Aberrant EGFR signaling also contributes to the dysregulation of the RAS/RAF/MEK/ERK cascade [30].

The molecular development of melanoma is mainly associated with mutations in the *v-Raf* (*BRAF*), *NF-1* (neurofibromin 1), *NRAS* (neuroblastoma RAS viral oncogene homolog), and *MC1R* (melanocortin 1 receptor) genes [31]. Over 50% of cells present the most frequent mutation of *v-Raf* gene, p.V600E; as a result, the hyperactivity of BRAF-MEK-ERK pathway is induced and subsequently leads to uncontrolled cell division [32]. BRAF activates MAPK-related MEKs, which subsequently stimulate ERK1/2 [33]. Other dysregulated pathways in melanoma include PI3K/AKT/mTOR and JAK/STAT [31]. The second most frequent mutations affect RAS-family GTPase genes, including NRAS [34]. The RAS/RAF/MAPK and PI3K/AKT signaling cascades are overactive in a constitutive manner, similarly to the BRAF mutation [35]. *NF1* gene mutations inactivate neurofibromin 1, a negative RAS regulator, leading to sustained MAPK and PI3K pathway activity [36]. Point mutations in *MC1R* impair α-MSH signaling, reducing the synthesis of eumelanin, a key UV-protective factor [37,38,39]. A simplified scheme presenting the molecular basis of the cellular pathways associated with melanoma development is depicted in Figure 5.

## 2. Berberine and Its Anticancer Activity

Berberine (BBR) is a substance widely used in traditional medicine, especially Chinese and Ayurvedic medicine, but its properties are also a subject of interest for conventional medicine. This substance has been vastly investigated in recent years, e.g., in the treatment of diabetes, Alzheimer’s disease, cardiovascular diseases, and cancer [40]. BBR is a benzylisoquinoline alkaloid (Figure 6) that naturally occurs in some plants, namely, the family *Berberidaceae* and its representative *Berberis vulgaris*. Its effects on human health are multidimensional, including changes in metabolic parameters (glycemia, insulin resistance, blood lipids, and body composition) and inflammatory markers, and activity against colorectal adenomas and infections caused by *Helicobacter pylori* [41]. As for its anticancer activity, current research suggests that BBR is potentially useful against different types of neoplasms, including pancreatic cancer [42], gastric cancer [43], lung cancer [44], colorectal cancer [45], breast cancer [46], and astrocytoma [47].

### 2.1. Anticancer Effects of Berberine

Studies propose many pathways in which the substance can exert an anticancer effect, acting both in stages before cancer formation as prophylaxis and in treatment of pre-invasive and invasive neoplasms. According to systematic review by Huang et al., BBR acts through the inhibition of cell proliferation, promotes apoptosis and autophagy in cancer cells, and prevents metastasis and angiogenesis [48]. The authors claim that the underlying mechanism mainly involves regulating signaling by activating adenosine monophosphate-activated protein kinase (AMPK) and inhibiting phosphatidylinositol 3-kinase/protein kinase B (PI3K/AKT). They also indicate the importance of BBR in causing the activation of forkhead box transcription factor O3a (FOXO3a), accumulation of reactive oxygen species (ROS), and inhibition of mTOR and nuclear factor-κB (NF-κB). Regulation of the PI3K/AKT pathway is also one of the possible mechanisms of BBR activity mentioned in review by Kaboli et al. [49], alongside other cell signaling pathways. Moreover, the mentioned mTOR/ PI3K/AKT pathway plays an important role in the case of skin cancers, both melanoma and non-melanoma, as a prognostic factor and therapeutic target [50]. Aberrances in this pathway were identified in various human cancer cells, including mutations in PI3K and mutations/amplifications of AKT [51]. Heightened AKT/mTOR activity occurs in about 70% of metastatic melanoma and it is reported that AKT activation is associated with a change from the radial growth phase of melanoma to the vertical growth phase [50]. Due to that, both synthetic and natural compounds inhibiting this pathway are being studied, e.g., curcumin, silymarin, and fisetin [52]. The inhibition of this pathway as a possible therapeutic mechanism was also suggested for BBR in cancers such as gastric cancer [53], colorectal cancer [54], and non-small-cell lung cancer [55]. For example, in the case of gastric cancer, the inhibition of the AKT pathway by BBR, alongside the upregulation of Beclin-1 and microtubule-associated protein 1 light chain 3, resulted in cytostatic autophagy both in vitro and in vivo [53]. The simplified anticancer effect of BBR is schematically depicted in Figure 7. 

Other mechanisms suggested by Kaboli et al. involve the inhibition of gene activators, such as NF-κB, growth factor receptors, e.g., epithelial growth factor receptor (EGFR), and apoptosis inhibitors (e.g., Bcl2) [49]. According to this study, BBR promotes the activity of tumor suppressors such as p53 and apoptosis induction pathways, among others, through caspases: caspase 3, caspase 8, and caspase 9. The authors suggest that also microRNAs may play a role, but the effects of BBR on microRNAs related to cancers have not been explored enough. However, since then, other studies explored the function of miRNAs in cancer suppression by BBR. In the colon cancer cell line HCT116, BBR inhibited miR-21 expression, promoting the expression of the ITGβ4 and PDCD4 proteins [58]. In HepG2 hepatoma cells, BBR increased miR-21-3p expression, which in turn reduced the expression of methionine adenosyl transferase (MAT), suppressing growth and inducing apoptosis [59]. One more mentioned pathway includes reactive oxygen species (ROS), the production of which is promoted by BBR. They result in the release of cathepsin B and apoptosis inhibitory factor (AIF) and the induction of apoptosis [60,61]. This mechanism was also used in combination with photodynamic therapy in renal cell carcinoma, as BBR is also a photosensitive agent that is capable of generating ROS in the presence of light, Therefore, the substance can be used as a photosensitizer [62]. The proposed mechanisms of BBR’s anticancer activity are summarized in Table 2.

The extensive pharmacological activity of BBR against cancer is supported by the evidence presented by Zhang et al. The authors highlighted the compound’s effect on the cell cycle by regulating the expression of p53 [53]. Moreover, this review emphasizes BBR’s ability to enhance the sensitivity of tumor cells to chemotherapeutic agents while simultaneously mitigating treatment-associated side effects. BBR also functions as an inhibitor of nuclear factor erythroid 2-related factor 2 (NRF2), a transcription factor that is frequently hyperactivated in cancer cells and implicated in resistance to anticancer therapies [63].

The role of BBR in modulating chemoresistance mechanisms is further explored by Sajeev et al. The authors identify key actions, including DNA damage induction, inhibition of efflux pumps, downregulation of multidrug resistance (MDR) genes, and the regulation of signaling pathways such as EGFR, NF-κB, angiogenesis suppression, and apoptosis modulation [64]. Notably, the authors propose BBR as a potential chemosensitizer when used in combination with agents such as cisplatin, doxorubicin, lapatinib, tamoxifen, irinotecan, or niraparib.

BBR anticancer activity was also tested in in vivo studies. Li et al. used subcutaneous xenograft tumor model mice. In this study, 39 mg/kg of BBR was administered intragastrically every day, which resulted in reductions in the tumor volume and weight compared to the controlled group. Additionally Ki-67 expression was reduced and β-catenin was suppressed [65]. Xenograft model mice were also used in a study by Huang et al. The study examined effects of BBR and probiotics on the colon cancer cell line HT29 inoculated subcutaneously.BBR alone and with probiotics had similar effects on inhibiting tumor growth; however, the authors suggest that BBR can act on the gut microbiome and enhance sodium butyrate production, which can have anticancer effects [66]. A xenograft model was also used in the study of non-small-cell lung carcinoma (NSCLC) by Chen et al. In this case, 25 mg/kg of BBR was administered daily. The effect of the substance was the suppression of tumor growth and extensive necrosis, as seen in a histological examination of tumors [67]. Therefore, there is evidence from in vitro studies suggesting that BBR has anticancer activity; however, the utilization of other methods than xenograft mouse models is needed to confirm those results.

In summary, the anticancer activity of BBR is mediated through multiple molecular mechanisms, leading to increased cell death and DNA repair, as well as decreased cell cycle progression, DNA integrity, and tumor angiogenesis. Additionally, BBR modulates the transcription of genes involved in cancer progression, supporting its potential use as a stand-alone therapeutic or as an adjunct to enhance the efficacy of conventional chemotherapeutics.

### 2.2. Anticancer Effects of Berberine Derivatives

Chemical modifications of the BBR molecule can be developed to enhance the anticancer activity or pharmacological properties. One of the natural metabolites of berberine found in plants is berberrubine, which is more lipophilic and has greater affinity for the P-gp receptor, resulting in better bioavailability [68]. Both BBR and berberrubine can be further modified to increase bioavailability, e.g., by the addition of lipophilic groups. For example, in a study by Chang et al., six derivatives were examined for activity against NSCLC. In this in vitro examination, two molecules showed greater efficacy than simple BBR: 9-O-decylberberrubine bromide and 9-O-dodecylberberrubine bromide [69]. Their mechanism of action included the induction of cell cycle arrest, inhibition of tumorigenesis, and modulation of autophagy through the suppression of autophagic flux. The study by Ortiz et al. suggested that potent anticancer agents can be found among arylalkyl derivatives of BBR with aromatic groups in the C13 position [70]. In this study, colon cancer cell lines were suppressed by the examined derivatives more than by simple BBR, with the possible mechanisms including the induction of autophagy.

The literature on BBR derivatives covers a wide range of different cancer cell lines. Examples are shown in Table 3. However, skin cancers are omitted in the research. Therefore, a detailed description of their activities is beyond the scope of this article. Studies examining BBR derivatives’ activities against skin cancers should be encouraged, as it seems to be a promising field of study.

## 3. Berberine and Its Potential Anti-Melanoma Activity

### 3.1. Primary Cutaneous and Noncutaneous Melanomas: Origin and Dissemination

Melanoma is a malignant tumor originating from melanocytes, specialized cells responsible for producing melanin, the pigment located in the basal layer of the epidermis [75,76]. The origin of melanocytes lies in the neural crest, which explains why melanomas can also arise in noncutaneous sites, following the migration pathways of neural crest cells, such as the eye [77,78], mucosa (gastrointestinal [79,80] and genitourinary tracts [81,82]), and the meninges [83,84,85]. Figure 8 presents the embryonic development and differentiation of melanocytes. Cutaneous melanomas account for the vast majority (90%) of all melanomas, compared to the relatively rare primary noncutaneous melanomas, including ocular melanoma (<4%) and mucosal melanoma (<2%) [85]. It is worth emphasizing that noncutaneous melanomas are associated with significantly worse prognoses, as reflected by their lower overall 5-year survival rates [85,86]. Melanoma distinguishes itself from other skin cancers through its remarkable capacity for local, regional, and distant dissemination. Patients without clinical or histopathological evidence of lymph node involvement (e.g., following a negative sentinel lymph node biopsy) may nevertheless harbor distant metastases [87].

Given these challenges, the search for novel therapeutic strategies becomes critically important. One promising compound with potential anti-melanoma activity is BBR. Research indicates its multifaceted mechanisms of action against cancer cells, including melanoma, which involve processes such as the inhibition of proliferation [57] and induction of apoptosis [88].

### 3.2. In Vitro Studies

Metastasis is the primary cause of death among cancer patients [89,90]. Tumor dissemination plays a critical role in the formation of metastatic tumors and is a key factor behind the elevated mortality and morbidity associated with cancer. Therefore, metastasis constitutes the most significant threat to the lives of cancer patients, as it can sometimes evade detection by current technologies, thereby increasing the risk of disease progression [91].

Given that BBR blocks the migration and invasion of various cancer cells types (e.g., prostate and colorectal cancer cells) [92,93], Liu et al. investigated the effect of BBR on the molecular mechanisms involved in metastasis of human melanoma cells in vitro [94]. The results demonstrated that BBR inhibits the migration and invasion of human melanoma A375.S2 cells in vitro via the FAK, uPA, and NF-κB signaling pathways. Furthermore, A375.S2 melanoma cells resistant to PLX4032 (A375.S2/PLX) were also selected for the study. It was shown that BBR significantly inhibited the motility of A375.S2/PLX cells. PLX4032, also known as vemurafenib, is a potent, selective inhibitor that targets the mutated BRAF kinase, specifically blocking the ERK signaling pathway in cancer cells expressing the BRAF V600E mutation [94]. This mutation is highly relevant in clinical practice, as approximately half of patients with metastatic melanoma harbor the BRAF mutation, which leads to altered signaling and uncontrolled cell proliferation [95]. Furthermore, the investigation of vemurafenib-resistant cells was not incidental, as disease progression is observed in patients with the mutation during treatment, a phenomenon attributed to the resistance of kinases that modulate the signaling pathway. The causes of this resistance remain an unresolved subject of ongoing research [96,97].

The epithelial–mesenchymal transition (EMT) is a biological process where epithelial cells lose their characteristics and acquire mesenchymal traits, enabling migration and invasion, thus playing a key role in metastasis [98,99,100]. The concept of the EMT was addressed in a study investigating the effect of berberine (BBR) on the metastasis of B16 murine melanoma cells and the molecular pathways involved. It was demonstrated that BBR reduced the EMT and inhibited the metastasis of B16 melanoma cells. A plausible mechanism underlying berberine’s suppression of the EMT may be its ability to modulate the PI3K/AKT signaling pathway, deactivating its oncogenic activity typically observed in tumor progression. Subsequently, this deactivation resulted in a decrease in the expression of the retinoic acid receptor α (RARα) and an increase in the expression of retinoic acid receptor β and γ (RARβ and RARγ) [101]. The RAR nuclear receptors play multifaceted roles in regulating cellular proliferation, specialization, and viability [102]. Interestingly, the RARα/RARβ proteins were linked to PI3K/AKT, and this cross-interaction influenced the inhibition of EMT-associated proteins, thereby blocking the infiltration and dissemination of murine melanoma cells [101].

#### In Vitro Synergy of Combination Treatments

BBR demonstrates potential in combination therapies targeting melanoma cells. However, despite cisplatin’s (CisPt) broad use in oncology, melanoma cells exhibit resistance, limiting its efficacy in melanoma treatment [103,104]. Due to the need for effective combinations, an in vitro study examined BBR and CisPt co-treatment, highlighting chemo-photodynamic therapy’s potential to improve cancer treatment efficacy [105]. Berberine is a unique compound that can function as a potent photosensitizer in photodynamic therapy (PDT) due to its ability to absorb light and generate reactive oxygen species (ROS). This leads to oxidative damage, DNA structural disruption, and cell death [106]. The study utilized melanoma cells resistant to CisPt and examined the effects of combining cisplatin with berberine–photodynamic therapy (BBR-PDT). It was demonstrated that this PDT could serve as a therapeutic approach for melanoma, as this combination of anticancer agents could overcome tumor resistance to cisplatin. Notably, BBR-PDT pretreatment could promote melanoma cell responsiveness to cisplatin-induced apoptosis, a process facilitated by the activation of the ROS/p38/caspase cascade [105].

Another study conducted on human melanoma cells also described apoptosis induced by photodynamic therapy using BBR-PDT. During this therapy, BBR induced endoplasmic reticulum (ER) stress, caused by an increase in ROS levels. In turn, ER stress triggered the overexpression of the apoptosis-related protein CHOP. Therefore, BBR-PDT affected CHOP, which was involved in the ER–stress–autophagy pathway, stimulating the death of human melanoma cells. These findings may suggest the potential of this approach as a future treatment for melanoma [88].

A different synergistic combination of therapeutic agents is berberine chloride and evodiamine (EVO), which exhibit suppressive effects on multiple cancer cell lines [57,107]. Evodiamine is a recently identified alkaloid that limits the proliferation of melanoma cells resistant to vemurafenib through the disruption of the PI3K/AKT signaling pathway [108]. Initially, the delivery of BBR and EVO to the skin posed a challenge, which led to the development of an ethosome formulation, optimizing the penetration of the substances into the epidermis. Appropriately designed ethosomes demonstrated an inhibitory effect on B16 melanoma cells, confirming that the combination of BBR and EVO in ethosomes may serve as a viable administration mechanism for these alkaloids, with possible application in melanoma therapy [57].

### 3.3. In Vitro and In Vivo Studies

Due to limited chemotherapy efficacy in advanced melanoma, immune checkpoint inhibitors have significantly improved survival [76,109,110,111]. This type of checkpoint immunotherapy represents a class of modern anticancer drugs that disrupt the pathways tumors exploit to evade immune surveillance and destruction [112]. A study by Luo et al. demonstrated that BBR could offer therapeutic benefits in immune checkpoint blockade (ICB) therapy by amplifying immune cell responses. BBR enhanced intracellular ROS production by blocking quinone oxidoreductase 1 (NQO1) activity to some extent [113]. NQO1 is an antioxidant enzyme regulated by the transcription factor NRF2, exhibiting elevated activity under the conditions typical of human malignant tumors. The expression of NQO1 in tumors is significantly higher compared to other tissues, establishing NQO1 as a selective biomarker for cancer [114]. The suppression of NQO1 triggered immunogenic cell death in melanoma cells, leading to the mobilization of dendritic cells (DCs) and CD8+ T lymphocytes both in vitro and in vivo, thereby serving as a pivotal factor in the anti-tumor immune response [113].

Another analysis presented berberine’s action differently. In this study, the in vivo organisms used were Danio rerio (zebrafish), a widely utilized animal model due to its ease of breeding and the presence of numerous human genes linked to diseases [115,116]. Furthermore, extracts from various Berberis species and their main alkaloids, BBR and palmatine, were applied. Palmatine demonstrates antimetastatic, anticancer, and antiproliferative effects [117]. The most significant cytotoxicity against the human melanoma cell line (A375) in vitro was observed for the extract obtained from *Berberispruinosa*. In contrast, in an in vivo experiment using A375 cells xenografted into zebrafish embryos, a decrease in the number of cancer cells was identified. Both in vitro and in vivo investigations confirmed the remarkable cytotoxic activity of the mentioned extract [118].

#### In Vitro and In Vivo Synergy of Combination Treatments

The synergistic interaction of BBR with other drugs has also been investigated in experiments with doxorubicin (DOX), a drug that, similar to CisPt, shows limited effectiveness in treating melanoma [103,119]. Studies conducted on rats suggest the potential nephroprotective, cardioprotective, and neuroprotective effects of BBR when combined with DOX, effectively mitigating its adverse effects [120,121,122]. The combination of BBR and DOX was applied to murine melanoma cells in culture as well as in a tumor xenograft model in mice. BBR potentiates the action of DOX, collectively suppressing cell expansion and division and promoting apoptosis in B16F10 melanoma cells, both in vitro and in the xenograft model. It is also worth noting that BBR attenuated DOX-induced toxicity in mice. Therefore, the combination of BBR and DOX represents a promising treatment strategy for melanoma [123].

Another example of combination therapy is photothermal therapy (PTT), a minimally invasive treatment modality that generally employs photothermal agents activated by laser irradiation to induce localized thermal effects [124,125]. Nanomaterial-based approaches in PTT are extensively utilized as ablative treatment modalities for managing a wide range of malignancies [126]. Polydopamine serves as an example of such nanoparticles that are capable of converting near-infrared (NIR) light into thermal energy, thereby facilitating the targeted interaction with cancer cells [127]. Ding et al. synthesized a composite by conjugating polydopamine (MPDA) with BBR, resulting in the formation of BBR-MPDA [128]. This conjugation proved highly advantageous, as the MPDA structure enabled the efficient loading and delivery of a substantial amount of BBR, leading to the effective ablation of cancer cells. The therapeutic efficacy of BBR-MPDA was evaluated both in vitro and in vivo using murine models bearing B16F10 melanoma tumors, where the combination therapy markedly outperformed the monotherapy in inhibiting tumor growth [128].These results highlight the potential of BBR-MPDA as an effective agent for improved photothermal cancer therapy.

To summarize, BBR demonstrates potential as a chemotherapeutic agent in melanoma therapy; however, further studies are required, particularly under in vivo conditions, which remain insufficiently explored. Complementing the existing findings would provide a deeper understanding of its mechanisms of action and allow for the assessment of its efficacy and safety in the clinical context of melanoma treatment.

A summary of the impact of BBR on melanoma cells is presented in Figure 9.

## 4. Berberine’s Role in the Treatment and Prevention of Cutaneous Squamous Cell Carcinoma

### 4.1. Characteristics of Cutaneous Squamous Cell Carcinoma

Cutaneous squamous cell carcinoma (SCC) takes second place in prevalence among all skin cancers [129]. It originates from the spinous layer of the epidermis and can appear on every part of the skin [130]. Histologically there are 11 differentiated subtypes of SCC [131], which include fi Bowen’s disease, keratoacanthoma, and invasive SCC [132]. Depending on the subtype and the tumor size, the survival rate and chances of metastates vary. One prospective study showed that lesions under 2 mm tend to not metastasize at all, while those over 16 mm have a 16% chance of metastasis [133]. It has been also evidenced that desmoplastic and adenosquamous SCC are characterized by a poorer course of the disease [23]. The overall 3-year survival rate for patients diagnosed with this cancer, without metastasis, remains high. For individuals with a low risk of metastasis, the rate is 91.4%; for those with a high risk, it drops to 80.6%; and for patients at the highest risk, it decreases further to 44.0% [134].

Currently known methods of treating SCC are invasive and carry risks of complications. This highlights the need to search for new, less risky treatment options. Among the promising candidates is berberine (BBR), which has shown remarkable antiproliferative properties in both in vitro and in vivo studies, effectively preventing metastasis.

### 4.2. In Vitro Studies

In vitro studies conducted on BBR and SCC have shown very promising results. One of them was performed in 2015 on the A431 cell line, which has been isolated from SCC [135]. The authors examined the effect that berberine has on those cancer cells. Through various tests such as the trypan blue dye exclusion assay or wound healing assay, it has been revealed that BBR inhibits the proliferation and migration of cells and it appears to promote cancer cell death.

An analysis performed by Li et al. showed that BBR increased the BAX/BCL-2 ratio. Those proteins belong to BCL-2 family and control the process of intrinsic apoptosis. In various tumors, the expression of the anti-apoptotic BCL-2 molecule is significantly increased [136], which may lead to an invulnerability to apoptosis signals. This protein, however, can be inactivated by its pro-apoptotic counterpart, the BAX protein [137]. It can be activated due to metabolic stress, hypoxia or DNA damage via BH3 only proteins [138]. Increasing the BAX-to-BCL2 ratio stimulates mitochondria to release cytochrome-c into cytosol, which activates the caspase cascade, thus leading to apoptosis [139]. This may explain why A431 cells in the berberine-treated group exhibited lower viability compared to the cells in the control group in this study. Figure 10 presents the relation between BBR and intrinsic apoptosis.

In this research, it has also been observed that BBR reduced the expression of the Ezrin protein. This molecule plays a crucial role in the cell membrane as it acts in many ways, such as linking the actin cytoskeleton to the extracellular matrix or surface receptors on the cell [140]. In cancer cells, Ezrin expression is elevated, which worsens the patient’s prognosis by facilitating tumor metastasis [53,141]. This may be caused by its interactions with proteins known to influence metastasis, including CDH1, CD44, and ICAM-1 [142]. Ezrin expression is also critical in SCC, as it is significantly higher in mutated cells compared to the healthy ones [143] and its interactions with CD44 can promote its progression easily [144]. A berberine-induced decrease in Ezrin expression could explain the inhibition of the migration of SCC cells in the study.

The anticancer effects of BBR in combination with other substances have also been studied. Extracts from *Argemone Mexicana* leaves have been shown to contain BBR along with other substances, e.g., protomexicine or pancorine. A 2024 study conducted by Kulshrestha et al. analyzed its antineoplastic properties [145]. It has been tested on two cell lines, one them being A431 cell line isolated from SCC. In this research, the use of the extract reduced SCC cell proliferation by 67%.

The predicted anti-SCC mechanism is linked to an observed decrease in the expression of TNF-α and NF-κB. TNF-α is a pro-inflammatory protein that activates NF-κB signaling, which is responsible for inflammatory response [146]. This pathway’s pathologies may lead to autoimmune conditions like skin psoriasis or cancers like SCC [147]. NF-κB in tumors stimulates autophagy [148], which proved to be one of mechanism behind cancer treatment resistance [149]. This may be the explanation for the antiproliferative features of the *Argemone Mexicana* leaf extract. Its enrichment in BBR and other bioactive substances gives it anti-SCC properties.

The form of the medication also plays an important role, as BBR has poor water solubility and low absorption in the gastrointestinal tract [150]. Another study from 2024 conducted by Solanki et al. compared the antineoplastic activity of BBR and sialic acid receptor targeted 4-carboxyphenylboronic acid-modified pullulan–stearic acid conjugate (4-cPBA-PUL-SA) filled with BBR (BPPNPs) on the A431 cell line isolated from SCC [151]. It has been observed that those BPPNPs exhibited superior antiproliferative properties compared to free BBR, as lower cell viability could be achieved using the nanoparticles at lower concentrations in comparison to the pure substance. The BPPNPsIC_50_ value after 48 h is over 2 times higher than that of BBR and after 24 h, it is over 8 times higher. The analysis also showed its remarkable pro-apoptotic activity against SCC through a staining assay. This could mean that the anticancer properties of BBR described above can be even enhanced with a proper form. A summary of in vitro anticancer properties of berberine against melanoma and SCC is presented in Table 4.

### 4.3. In Vitro and In Vivo Studies

BBR’s antineoplastic properties have been also proven in a combined in vitro and in vivo study conducted in 2023, which compared the effectiveness of BBR and erlotinib alone and in combination on the A431 cell line and mice injected with tumor cells from this line [152]. Erlotinib belongs to the first generation of epidermal growth factor receptor (EGFR) tyrosine kinase inhibitors. It is highly effective against EGFR(+) cancers; however, due to the therapeutic dose being close to the upper limit of tolerance, its use may entail serious adverse effects [153]. The research proved that the addition of BBR to erlotinib significantly improved its anticancer activity without increasing its dose in both in vitro and in vivo experiments. Table 5 provides a summary of BBR drug combinations, along with the type of study and the cell lines employed.

The presumed molecular mechanism behind the effectiveness of BBR’s addition could be the decreased BCL-2/BAX ratio and increased dephosphorylation of EGFR, which renders it inactive. This claim has been proven through various assays conducted on both in vitro A431 cells and cells retrieved from mice in this research. BBR’s inhibition of EGFR has been also observed in other studies [64]. This receptor’s high expression seems to indicate a poor prognosis for SCC patients as it is linked to metastasis progression [154]. In last few years, EGFR inhibitors play ever growing roles in this cancer’s treatment [155], and so, BBR’s role as a potential non-toxic antineoplastic drug or addition to prejacent ones can grow and provide patients with a less straining and more effective therapy.

## 5. Conclusions

This review we presented here confirms the high anticancer potential of BBR against skin cancers. Previous studies have shown that this alkaloid can (i) inhibit tumor cell proliferation while at the same time inducing apoptosis; (ii) affect signaling pathways (e.g., MAPK/ERK and PI3K/Akt); and (iii) inhibit tumor cell migration and invasion (e.g., MMP-2 and MMP-9). In addition, BBR has also been shown to have a limiting effect on oxidative stress and chronic inflammation, the main factors responsible for the first stages of skin carcinogenesis. Furthermore, it is interesting to note that a synergistic effect of this alkaloid has been shown to promote a chemosensitizing effect with anticancer drugs, i.e., cisplatin, doxorubicin and vemurafenib. These results suggest the emergence of a new quality in antineoplastic treatment—one based on natural derivatives with limited side effects and the ability to inhibit chemotherapy resistance, in contrast to existing therapies. If proven safe in clinical trials, the introduction of BBR as an anticancer agent could reduce chemotherapy-related complications and improve survival outcomes. This study supports the potential of BBR and highlights the need for continued research. However, despite these promising findings, further comprehensive studies are still required to explore various aspects of its development and application.Future research should not only include the selection of an appropriate concentration of this alkaloid and form of application (e.g., creams and ointments) but also the developmentof more effective derivatives characterized by greater bioavailability. Particular attention should be focused on various types of carriers, such as nanoparticles that transport BBR directly to the tumor lesion. It is also crucial to include the effect of BBR on reducing the toxicity of chemotherapy and radiotherapy as it poses a great challenge in modern oncology. It is also important to acknowledge that there are two factors that may compromise the quality of our research. First, in vivo research on the use of BBR in skin cancers is limited and so to accurately assess its effects on living organisms, further studies are necessary. Second, there is a lack of recent publications on the topic, which may affect the reliability of our findings. Above all, an important aim of future research should be to understand more precisely the interaction of BBR with anticancer drugs, which would allow the design of more effective therapeutic strategies, including personalized ones.

## 6. Future Scope and Perspectives

However, these promising reports still require developmental studies covering a wide range of aspects. Future research should focus on the effect of BBR on reducing the toxicity of chemotherapy and radiotherapy, as well as possible negative interactions with current anticancer drugs and treatment strategies. Studies should determine the optimal concentrations of this alkaloid and forms of application, both external, e.g., creams and ointments, and systemic, such as oral or intravenous. Particular attention should focus on various types of carriers, such as nanoparticles, that transport BBR directly to the tumor lesion. Also, BBR’s chemical modifications should be investigated to find the most effective and safe derivatives. Above all, in vivo studies concentrated on BBR use in treatment of skin cancers are necessary.

## Figures and Tables

**Figure 1 cells-14-01041-f001:**
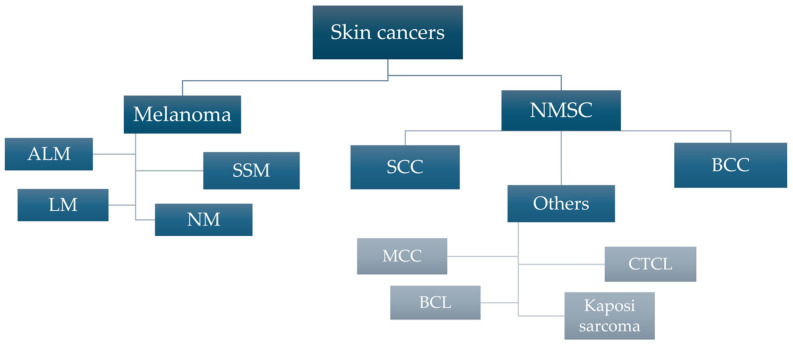
Classification of skin cancers with special regard to melanoma phenotypes and representatives of non-melanoma skin cancers (NMSC). Abbreviations: ALM—acral lentiginous melanoma, BCC—basal cell carcinoma, BCL—B-cell lymphoma, CTCL—cutaneous T-cell lymphoma, LM—lentigo melanoma, MCC—Merkel cell carcinoma, NM—nodular melanoma, NMSC—non-melanoma skin cancer, SCC—squamous cell carcinoma, SSM—superficial spreading melanoma.

**Figure 2 cells-14-01041-f002:**
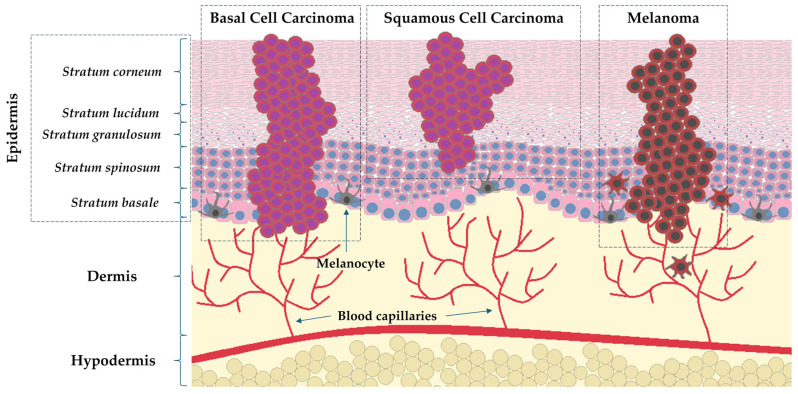
Micro-structure of the skin with a special regard to the histological image of the layers of epidermis and start points of skin cancer proliferation (simplified scheme).

**Figure 3 cells-14-01041-f003:**
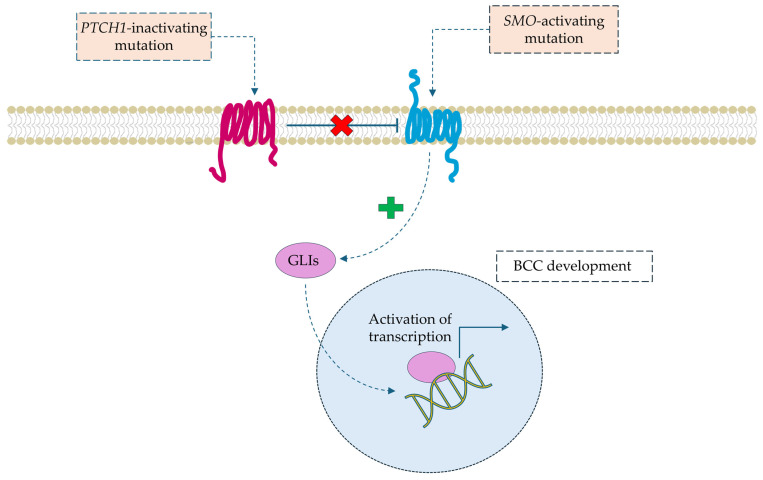
Overactivity of the Hedgehog signaling pathway associated with BCC development (simplified scheme). Abbreviations: BCC—basal cell carcinoma, GLIs—glioma-associated oncogenes, *PTCH1*—gene encoding the patched homolog 1 protein, *SMO*—Smoothened gene.

**Figure 4 cells-14-01041-f004:**
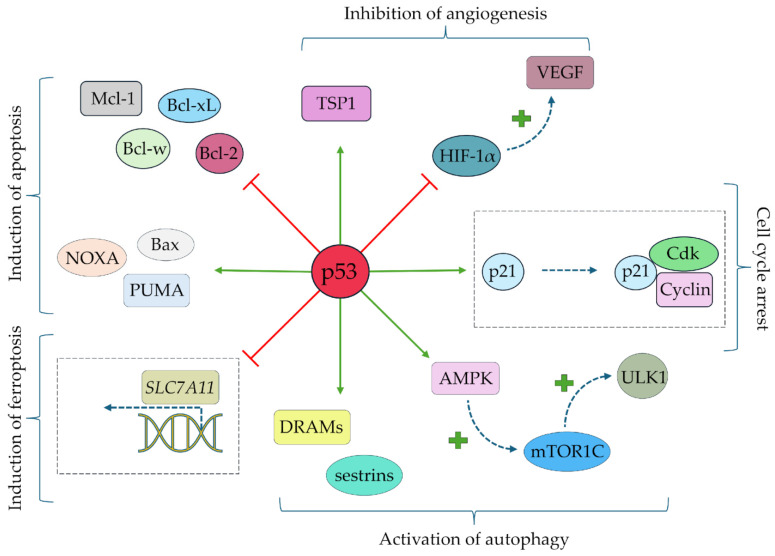
Molecular roles of selected *tp53*-mediated cellular pathways under normal conditions (simplified scheme). Abbreviations: AMPK—5′-adenosine monophosphate-activated protein kinase, Bax—apoptosis regulator BAX; Bcl-2, Bcl-w, and Bcl-xL—pro-apoptotic factors, Cdk—cyclin-dependent kinase, DRAMs—damage-regulated autophagy modulators, HIF-1α—hypoxia-inducible factor 1 α, Mcl-1—pro-apoptotic factor Mcl-1, mTOR1C—mammalian target of rapamycin 1C, NOXA—phorbol-12-myristate-13-acetate-induced protein 1, PUMA—p53 upregulated modulator of apoptosis, p21—cyclin-dependent kinase inhibitor 1, p53—tumor suppressor protein p53, *SCL7A11*—cystine/glutamate transporter-encoding gene, TSP1—thrombospondin 1, ULK1—serine/threonine-protein kinase ULK1.

**Figure 5 cells-14-01041-f005:**
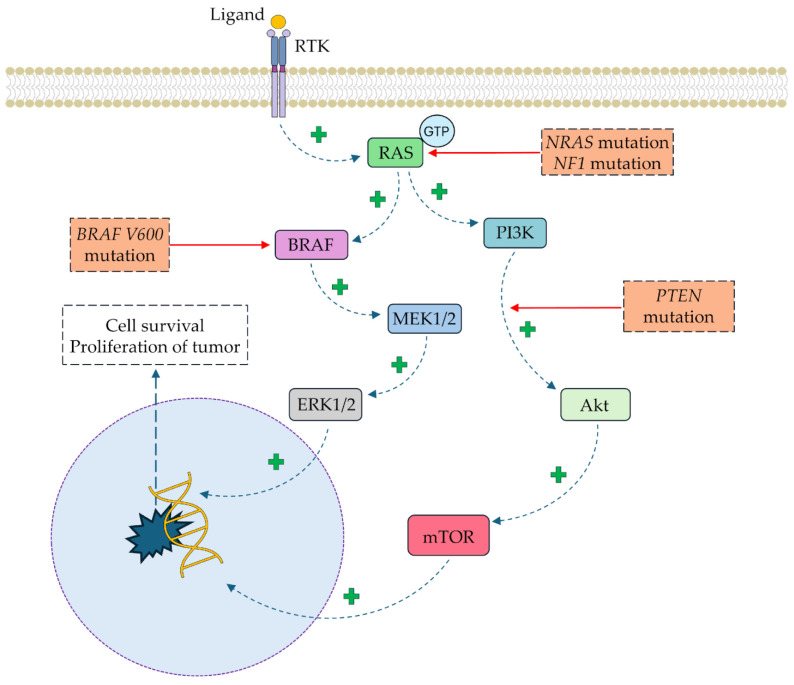
The main signaling pathways affected in melanoma. Abbreviations: Akt—protein kinase B, *BRAF*—*v-Raf* murine sarcoma viral oncogene homolog B, ERK—extracellular signal-regulated kinase, GTP—guanosine triphosphate, MEK—mitogen-activated protein kinase kinase, mTOR—mammalian target of rapamycin, *NF1*—neurofibromin 1, *NRAS*—neuroblastoma RAS viral oncogene homolog, *PTEN*—phosphatase and tensin homolog, RAS—rat sarcoma virus kinase, RTK—receptor tyrosine kinase.

**Figure 6 cells-14-01041-f006:**
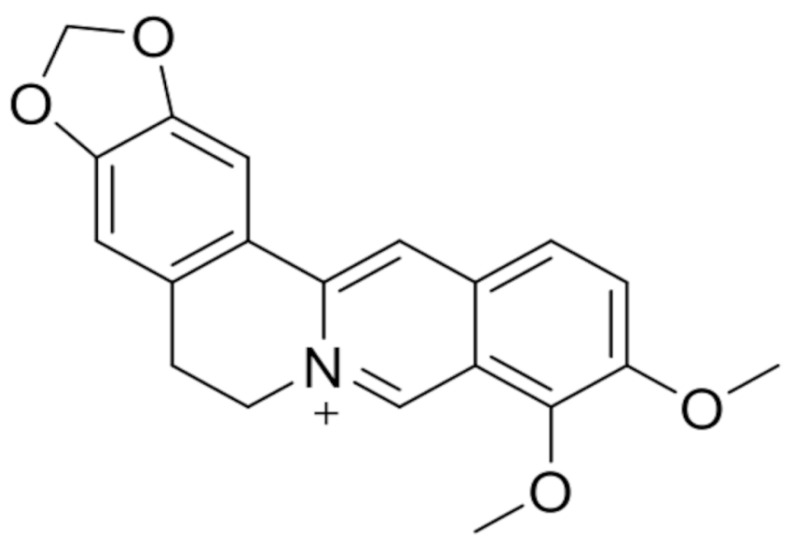
Molecular structure of berberine.

**Figure 7 cells-14-01041-f007:**
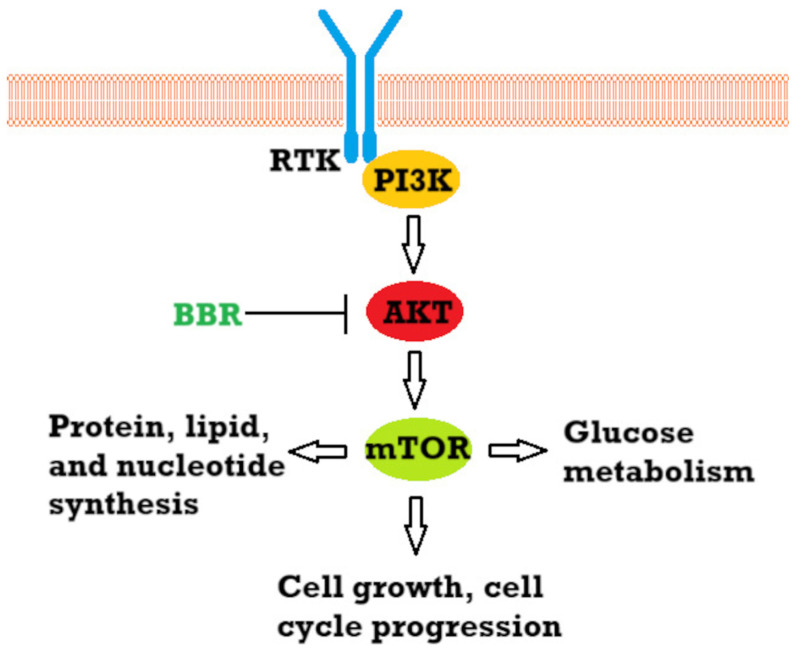
Simplified diagram of the mechanism of the anticancer activity of BBR through the inhibition of the PI3K/AKT/mTORpathway [50,53,56,57]. Abbreviations: Arrows represent activation, and bars indicate inhibition. RTK—receptor tyrosine kinase.

**Figure 8 cells-14-01041-f008:**
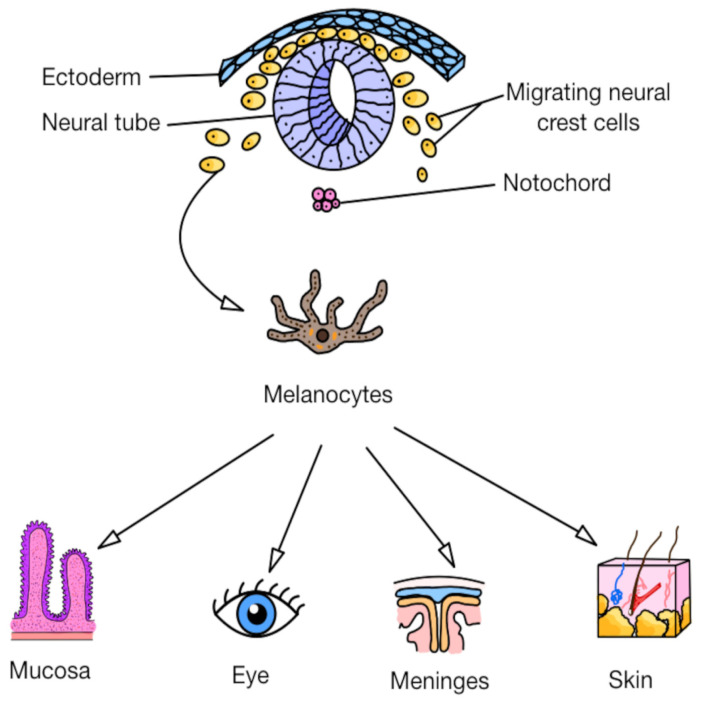
The embryonic development and differentiation of melanocytes.

**Figure 9 cells-14-01041-f009:**
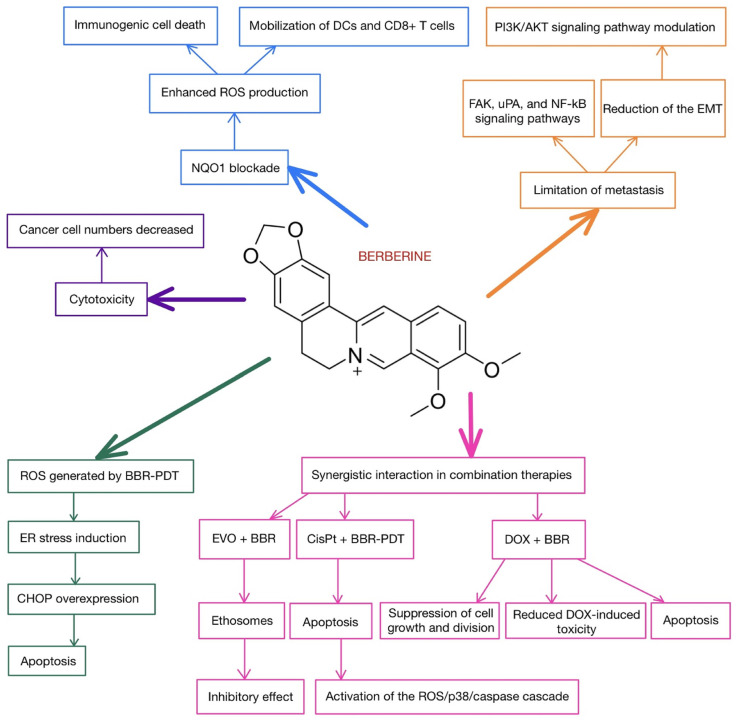
Spectrum of the molecular effects of berberine on melanoma cells. Abbreviations: EMT—epithelial–mesenchymal transition, CisPt—cisplatin, BBR-PDT—berberine–photodynamic therapy, ROS—reactive oxygen species, ER—endoplasmic reticulum, CHOP—apoptosis-related protein, EVO—evodiamine, BBR—berberine, DOX—doxorubicin, NQO1—quinone oxidoreductase 1, DCs—dendritic cells.

**Figure 10 cells-14-01041-f010:**
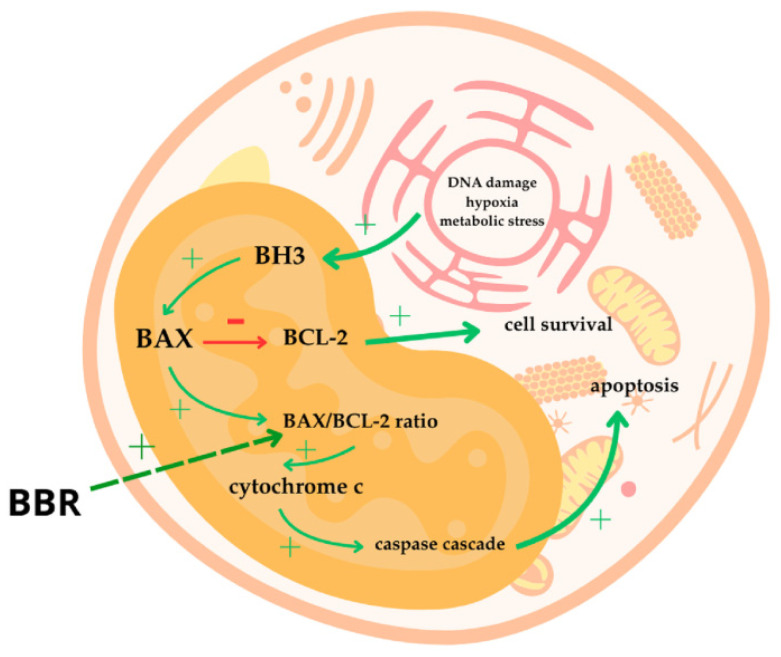
The relation between BBR and intrinsic apoptosis.

**Table 1 cells-14-01041-t001:** Comparison of the characteristics of the three main skin cancers.

	BCC	SCC	Melanoma
**Prevalence**	Most common skin cancer	2nd most common skin cancer	3rdmost common skin cancer
	80–90% of skin cancers 3.6 mln/annually (USA)	15% of skin cancers 1.8 mln/annually (USA)	5% ≥ of skin cancers 200,340 cases in 2024 (USA)
**Localization**	Sun-exposed areas, e.g., face, ears, scalp, neck, shoulders, back		Various locations, often not exposed to sunlight
**Riskfactors**	UV radiation	The main risk factors are identical to BCC	UV radiation
	History of BCC/SCC		Burns in childhood
	Age (median age of 68 years)	HPV 16/18/31/33/35 infections	Immunosuppression
	Male gender	Smoking	
	Fitzpatrick skin types I and II	Precancerous conditions, e.g., keratosis actinica and xeroderma pigmentosum Carcinoma in situ: Bowenoid papulosis and Erythroplasia of Queryrat	History of melanoma
	Immunosuppression		
	History of transplantation		Fitzpatrick skin types I and II
	Genetic conditions, e.g., Gorlin–Goltz syndrome		Genetic conditions: atypical nevi, nevi congenitales, and atypical nevus syndrome
**Growth** **characteristics**	Slow growth, causing minimal damage to the surrounding tissue	Rapid growth, may cause local tissue destruction (especially SCC exulcerans)	Depends on the type: slow growth of LM, SMM and ALM, and rapid growth of NM
**Metastases**	Rarely metastasizes, mainly to regional lymph nodes, bones, lungs and skin	More frequent than BCC, mainly to lymph nodes	High risk of metastases, main locations: subcutaneous tissue, regional lymph nodes, lungs, CNS, and liver
**References**	[6,7,8,9,10]	[6,8,11,12,13]	[6,14,15,16,17,18]

**Abbreviations**: ALM—acral lentiginous melanoma, BCC—basal cell carcinoma, CNS—central nervous system, HPV—human papilloma virus, LM—lentiginous melanoma, NM—nodular melanoma, SCC—squamous cell carcinoma, SSM—superficial spreading melanoma, UV—ultraviolet.

**Table 2 cells-14-01041-t002:** Possible mechanisms of BBR’s anticancer activity, as proposed by Kaboli et al. [49].

Mechanism of the Activity	BBR’s Effect	Molecules Involved
Cyclin/Cdk levels	Suppression	Cyclin D1
Growth ractor receptors	Suppression	EGFR, Her2/neu, PDGFR, VEGFR2
Gene activators	Suppression	AP1, AP2, NF-κB
Tumor suppressors	Activation	p21, p27, p53
Cell signaling pathways	Suppression	ERK/MAPK, PI3K/AKT, wnt/B-catenin
Apoptosis induction	Activation	Caspases, Bid, Bax, AIF, FasR, TNF-alpha
Apoptosis inhibition	Suppression	Bcl2, XIAP, IAP
microRNAs	Suppression/Activation	miR-21, miR-21-p

**Abbreviations**: AIF—apoptosis-inducing factor, AP1—activator protein 1, AP2—activator protein 2, Bax—Bcl-2-associated X protein, Bcl2—B-cell lymphoma 2 protein, Bid—BH3-interacting domain death agonist, CDK—cyclin-dependent kinase, EGFR—epithelial growth factor receptor, ERK/MAPK—extracellular signal-regulated kinase/mitogen-activated protein kinase pathway, FasR—Fas receptor, IAP—inhibitors of apoptosis, NF-κB—nuclear factor kappa-light-chain-enhancer of activated B cells, PDGFR—platelet-derived growth factor receptor, PI3K/AKT—phosphatidylinositol 3-kinase/protein kinase B pathway, TNF-alpha—tumor necrosis factor-alpha, VEGFR2—vascular endothelial growth factor receptor 2, XIAP—X-linked inhibitor of apoptosis protein.

**Table 3 cells-14-01041-t003:** Notable BBR derivatives with suggested activity against different cancer types.

BBR Derivatives with the Highest Anticancer Activity Shown	Type of Cancer	Reference
9-O-decylberberrubine bromide and 9-O-dodecylberberrubine bromide	Lung (NSCLC)	[69]
13-(4,4-diphenylbutyl)-9,10-dimethoxy-5,6-dihydrobenzo[g]-1,3-benzodioxolo[5,6-a]quinolizinium chloride		
13-(5,5-diphenylpentyl)-9,10-dimethoxy-5,6-dihydrobenzo[g]-1,3-benzodioxolo[5,6-a]quinolizinium chloride	Colon (HCT116 and SW613-B3)	[70]
9-O-dodecyl-, 13-dodecyl- and 13-O-dodecyl-berberine	Liver (HepG2)	[71]
13-[3-(phenyl)propyl]berberine iodide 13-[2-(4-cholorophenyl)ethyl]berberine	Breast (SK-BR-3)	[72]
9-O-farnesylberberine	Liver (HepG2)	[73]
9-O-6-ammonia chloride hexylberberine	Cervix (Siha), Lung (A549), Promyelocytic leukemia (HL-60)	[74]

**Table 4 cells-14-01041-t004:** The summary of in vitro activity of berberine against melanoma and squamous cell carcinoma.

Melanoma		
Actions of Berberine	Molecular Level	References
Reduced migration and invasion abilities	Downregulation of uPA Silencing of the FAK and NF-κB pathways	[94]
Reduced epithelial–mesenchymal transition	Downregulation of RARα/β Silencing of the PI3K/AKT pathway	[101]
Enhanced apoptosis rate of melanoma cells	Activation of the ROS-assisted p38/caspase pathway	[105]
Activation of autophagy and apoptosis	ROS-assisted endoplasmic reticulum stress Upregulation of the CHOP protein	[88]
Restoration of tumor immunogenicity	Downregulation of NQO1 expression	[113]
Squamous cell carcinoma		
Enhanced apoptosis rate of SCC cells	Upregulation of BAX Downregulation of BCL-2	[136]
Reduced migration and invasion abilities	Downregulation of the Ezrin protein	[140]
Suppression of tumor-promoting inflammation	Downregulation of TNF-α Silencing of the NF-κB pathway	[146]

**Abbreviations**: Bax—Bcl-2-associated X protein, Bcl2—B-cell lymphoma 2 protein, FAK—focal adhesion kinase, NF-κB—nuclear factor kappa-light-chain-enhancer of activated B cells, NQO1—NAD(P)H quinone dehydrogenase 1, PI3K/AKT—phosphatidylinositol 3-kinase/ protein kinase B pathway, RARα/β—retinoic acid receptor α/β, ROS—reactive oxygen species, SCC—squamous cell carcinoma, TNF-α—tumor necrosis factor-alpha, uPA—urokinase-type plasminogen activator.

**Table 5 cells-14-01041-t005:** BBR drug combinations.

Melanoma			
Drug Combinations	Type of Study	Cell Lines Tested	References
BBR photodynamic therapy + cisplatin	in vitro	A375 SK-Mel-19	[105]
BBR + evodiamine ethosome formulation	in vitro	B16	[57]
BBR + doxorubicin	in vitro/in vivo	B16F10	[123]
Squamous cell carcinoma			
BBR + erlotinib	in vitro/in vivo	A431	[152]

## Data Availability

The data is contained within the article.

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
