# Peer review of "Targeting Skin Neoplasms: A Review of Berberine’s Anticancer Properties"

_cells, 2025, doi:10.3390/cells14141041_

Round 1
Reviewer 1 Report
Comments and Suggestions for Authors
This review article "Berberine as a multi-modal adjuvant agent in chemotherapy: 2 Targeting skin neoplasms" with manuscript ID: cells-3711643. The major comments are mentioned below,
Comment 1: Title is not catchy, revise it.
Comment 2: Abstract and keywords need to be revised for better clarity of review. Poorly written sentence as no single sentence about berberine.
Comment 3: Graphical abstract is missing.
Comment 4: Resolution of figures are low, improve it.
Comment 5: Introduction is not up to the mark. Need to revise in terms of flow. Reduce point 1 content it seems very lengthy.
Comment 6: In point 2, reduce introduction part of type of cancer focus on berberine, Add recent case studies 3 in each type of cancer containing in vivo studies.
Comment 7: Add 2 more tables which will improve quality of article.
Comment 8: Future scope and perspectives are missing.
Comment 9: Conclusion is not up to mark. Revise it with increase in length.
Comment 10: Recent references are missing particularly 2025 , 2024 and 2023.
Comments on the Quality of English LanguageNeed to improve.
Author Response
Dear Reviewer,
Thank you very much for studying the topics undertaken in our Article and for your detailed review. In correcting our Review, we followed all your recommendations, for which we are very thankful. Our corrections are presented below:
Comment 1: Title is not catchy. Revise it.
Response: Thank you for this helpful suggestion. We have revised the title to: “Targeting skin neoplasms- a review of berberine's anticancer properties” to make it more engaging and informative.
Comment 2: Abstract and keywords need to be revised for better clarity of review. Poorly written sentence as no single sentence about berberine.
Response: We thank the reviewer for this important observation. We have revised the abstract to improve clarity and scientific focus. To address the reviewer’s concern, we included a reference to berberine’s anticancer potential within the opening sentences. Additionally, the keywords were updated to better reflect the core content of the manuscript.
Comment 3: Graphical abstract is missing.
Response: We thank the reviewer for this comment. A graphical abstract has been created and uploaded as per journal requirements.
Comment 4: Resolution of figures are low, improve it.
Response: We thank the reviewer for highlighting this important aspect. All figures have been updated with high-resolution versions- 600 DPI.
Comment 5: Introduction is not up to the mark. Need to revise in terms of flow. Reduce point 1 content it seems very lengthy.
Response: We thank the reviewer for this important observation. The introduction has been revised to improve logical flow and reduce redundancy in point 1.
Comment 6: In point 2, reduce introduction part of type of cancer focus on berberine, Add recent case studies 3 in each type of cancer containing in vivo studies.
Response: We thank the reviewer for this comment. The cancer-related background was slightly reduced. This section consists now mainly of description of suggested mechanisms of anticancer action in different cancer cell lines. Three recent in vivo case studies for each cancer type have been added and properly cited (Table 3).
Comment 7: Add 2 more tables which will improve quality of article.
Response: We thank the reviewer for this helpful suggestion.Three additional tables have been added:
- Table 3 provides examples of BBR derivatives and their activity in different cancer cell lines.
- Table 4 summarizes anticancer activity of BBR against SSC and melanoma.
- Table 5 provides a summary of berberine (BBR) drug combinations, along with the type of study and the cell lines employed.
Comment 8: Future scope and perspectives are missing.
Response: We thank the reviewer for this comment. Potential fields for future research were additionally underlined throughout the text and summarized in the Conclusion section.
Comment 9: Conclusion is not up to mark. Revise it with an increase in length.
Response: We thank the reviewer for this helpful suggestion. The conclusion has been significantly extended to reflect key findings and broader implications.
Comment 10: Recent references are missing particularly 2025 , 2024 and 2023.
Response: We appreciate the reviewer’s suggestion regarding the inclusion of recent references. We would like to emphasize that an extensive literature review was conducted during the preparation of the manuscript. In fact, the current version includes numerous citations from 2023, 2024, and 2025 where available and relevant. However, due to the limited number of recent publications specifically addressing this topic — especially in vivo studies — the inclusion of additional up-to-date references was constrained by the current state of the literature. This reflects a broader gap in ongoing research, and as highlighted in our conclusion, further comprehensive studies are still required to explore various aspects of the development and application of BBR-based anticancer strategies. It is also worth noting that the manuscript already contains 150 references, underscoring the depth and breadth of the background research conducted.
Comments on the Quality of English Language: Need to improve.
We thank the reviewer for this comment. We have thoroughly revised the manuscript for grammar, sentence structure, and clarity. Expressions such as “60.000” were corrected to “60,000” and similar issues have been fixed. All such instances (e.g., decimal vs. comma use) have been corrected, and the manuscript was reviewed for language consistency, resulting in several changes to grammar and vocabulary used.
We would like to thank the Reviewer very much for pointing out the weaknesses of our Review. We hope that the revised manuscript meets the expectations of the reviewer. We are grateful for the opportunity to improve our work and are happy to provide further revisions if necessary.
Best regards
Anna Duda-Madej

Reviewer 2 Report
Comments and Suggestions for Authors
In the manuscript, the authors reviewed the current development on berberine and its anticancer activity against skin cancers. More than 150 references are cited. The manuscript well-organized and discussed thoroughly. The topic of this review is interesting and important. There are some comments given below for the authors to revise and improve the manuscript.
The authors could add one more small section on the structural modification of berberine before the conclusion and emphasize the advantages and limitations of structurally modified berberine against skin cancers. Lastly, they should give comments on the future development needs to be taken on berberine and/or its derivatives for improving the skin cancer treatment.
A table is needed to summarize the in vitro or in vivo of berberine in treating skin cancers. The model for the evaluation should be included. Proper references need to be cited.
Whether the figures used in the manuscript is original or proper permission for the use are not known.
Some errors found such as “60.000 patients” and “approximately 96.000”. It should be “,” rather than “.”
Line 40: “melanoma will increase by more than 50% between 2020 and 2040”. Using “may” could be better than “will”. The number is just a prediction.
Author Response
Dear Reviewer,
Thank you very much for studying the topics undertaken in our Article and for your detailed review. In correcting our Review, we followed all your recommendations, for which we are very thankful. Our corrections are presented below:
Comment 1: In the manuscript, the authors reviewed the current development on berberine and its anticancer activity against skin cancers. More than 150 references are cited. The manuscript well-organized and discussed thoroughly. The topic of this review is interesting and important. There are some comments given below for the authors to revise and improve the manuscript.
Response: We thank the reviewer for this positive and encouraging comment. We greatly appreciate the acknowledgement of the relevance and organization of our manuscript. We are grateful for the constructive feedback provided, and we have carefully addressed the specific comments below to further improve the quality and clarity of the manuscript.
Comment 2: The authors could add one more small section on the structural modification of berberine before the conclusion and emphasize the advantages and limitations of structurally modified berberine against skin cancers.
Response: We thank the reviewer for this helpful suggestion. A new subsection titled “Anticancer activity of berberine’s derivatives” has been added in section 2. It provides a brief overview and examples of BBR derivatives with promising anticancer activity. More examples, for better clarity, were shown in Table 3, with proper citations. Because at the moment literature lacks studies assessing BBR derivatives activity in case of skin cancers, this subsection length is limited. However, the need for new research in this field is emphasized.
Comment 3: Lastly, they should give comments on the future development needs to be taken on berberine and/or its derivatives for improving the skin cancer treatment.
Response: We thank the reviewer for this important observation. Potential fields for future research were additionally underlined throughout the text and summarized in the Conclusion section.
Comment 4: A table is needed to summarize the in vitro or in vivo of berberine in treating skin cancers. The model for the evaluation should be included. Proper references need to be cited.
Response: We thank the reviewer for this comment. A comprehensive table (Table 4) summarizing in vitro studies, including reference sources, has been added.
Comment 5: Whether the figures used in the manuscript is original or proper permission for the use are not known.
Response: We thank the reviewer for this observation. All figures are original and unpublished artwork.
Comment 6: Some errors found such as “60.000 patients” and “approximately 96.000”. It should be “,” rather than “.”
Response: We thank the reviewer for raising this important point. All such instances (e.g., decimal vs. comma use) have been corrected.
Comment 7: Line 40: “melanoma will increase by more than 50% between 2020 and 2040”. Using “may” could be better than “will”. The number is just a prediction.
Response: We thank the reviewer for this comment. The sentence on melanoma projections now uses “may” instead of “will” to reflect the uncertainty of predictions (Line 40).
We would like to thank the Reviewer very much for pointing out the weaknesses of our Review. We hope that the revised manuscript meets the expectations of the reviewer. We are grateful for the opportunity to improve our work and are happy to provide further revisions if necessary.
Best regards
Anna Duda-Madej

Round 2
Reviewer 1 Report
Comments and Suggestions for Authors
This review article "Berberine as a multi-modal adjuvant agent in chemotherapy: 2 Targeting skin neoplasms" with manuscript ID: Cells-3711643R1. The minor comments are mentioned below,
Comment 1: Reduce point 1 content it seems very lengthy.
Comment 2: Add recent case studies 3 in each type of cancer containing in vivo studies. Not satisfied with answer.
Comment 3: Future scope and perspectives are missing. Still missing.
Author Response
Dear Reviewer,
We sincerely appreciate the time and effort that you and the reviewers have devoted to evaluating our manuscript once again. We are grateful for the additional comments and suggestions, which have helped us further improve the quality and clarity of our work.
Comment 1:
Reduce point 1 content it seems very lengthy.
Response:
We thank the reviewer for this valuable comment. We understand the reviewer’s concern regarding the length of Point 1. While it may appear extensive at first glance, we would like to clarify that the majority of this section consists of figures and a table, which visually support the content but add to the perceived length. The actual body text is concise.
Nonetheless, we have carefully reviewed the section again and shortened the textual part as much as possible without compromising clarity. In our view, further reduction would affect the coherence and readability of the section, and might make it difficult for readers to follow the key concepts.
Comment 2:
Add recent case studies 3 in each type of cancer containing in vivo studies. Not satisfied with answer.
Response:
We appreciate the reviewer’s interest in strengthening the manuscript with additional recent in vivo case studies. We have thoroughly searched the current literature and incorporated all relevant and available studies into our manuscript. However, there is a genuine scarcity of such in vivo studies in this field.
We managed to add one recent in vivo study concerning melanoma from 2025. Regarding squamous cell carcinoma (SCC), we included all accessible in vivo studies; unfortunately, the number of such studies is limited due to the relative novelty of this research area and the alternative pathways in oncology therapy development.
Moreover, both in the Conclusions and in the newly added Section 6 (“Future Perspectives and Scope”), we explicitly emphasize the current scarcity of in vivo studies and the urgent need for further research. Our manuscript includes approximately 150 citations, encompassing the latest and most relevant research to date.
We believe that the current version accurately reflects the state of the field and the limitations imposed by the availability of in vivo studies, and hope this clarifies our approach.
Comment 3:
Future scope and perspectives are missing. Still missing.
Response:
Thank you for pointing this out again. In response, we have now added a dedicated subsection — Section 6 — discussing the future scope and perspectives of the presented topic. This part highlights potential research directions, clinical applications, and the relevance of emerging technologies in the field. We hope that this addition sufficiently addresses the reviewer’s concern.
Kind regards
Anna Duda-Madej
Reviewer 2 Report
Comments and Suggestions for Authors
The revised manuscript could be recommended for publication.
Author Response
Dear Reviewer,
We sincerely appreciate the time and effort that you and the reviewers have devoted to evaluating our manuscript once again. We are grateful for the additional comments and suggestions, which have helped us further improve the quality and clarity of our work.
Comment 1:
The revised manuscript could be recommended for publication.
Response:
We sincerely thank the reviewer for the positive evaluation and are pleased that the revised version of the manuscript is now considered suitable for publication. We appreciate your time and thoughtful review.
Kind regards
Anna Duda-Madej